# Detailed Three-Dimensional Building Façade Reconstruction: A Review on Applications, Data and Technologies

Anna Klimkowska [1,*], Stefano Cavazzi [2], Richard Leach[3] and Stephen Grebby [1]

1   Nottingham Geospatial Institute, University of Nottingham, Jubilee Campus, Nottingham NG7 2TU, UK;
    stephen.grebby@nottingham.ac.uk
2   Ordnance Survey, Adanac Drive, Southampton SO16 0AS, UK; stefano.cavazzi@os.uk
3   Department of Mechanical, Materials and Manufacturing Engineering, University of Nottingham,
    University Park, Nottingham NG7 2RD, UK; richard.leach@nottingham.ac.uk
*   Correspondence: anna.klimkowska@nottingham.ac.uk

**Abstract:** Urban environments are regions of complex and diverse architecture. Their reconstruction and representation as three-dimensional city models have attracted the attention of many researchers and industry specialists, as they increasingly recognise the potential for new applications requiring detailed building models. Nevertheless, despite being investigated for a few decades, the comprehensive reconstruction of buildings remains a challenging task. While there is a considerable body of literature on this topic, including several systematic reviews summarising ways of acquiring and reconstructing coarse building structures, there is a paucity of in-depth research on the detection and reconstruction of façade openings (i.e., windows and doors). In this review, we provide an overview of emerging applications, data acquisition and processing techniques for building façade reconstruction, emphasising building opening detection. The use of traditional technologies from terrestrial and aerial platforms, along with emerging approaches, such as mobile phones and volunteered geography information, is discussed. The current status of approaches for opening detection is then examined in detail, separated into methods for three-dimensional and two-dimensional data. Based on the review, it is clear that a key limitation associated with façade reconstruction is process automation and the need for user intervention. Another limitation is the incompleteness of the data due to occlusion, which can be reduced by data fusion. In addition, the lack of available diverse benchmark datasets and further investigation into deep-learning methods for façade openings extraction present crucial opportunities for future research.

**Keywords:** façade parsing; building openings; object detection; images; point cloud; platforms; sensors

## 1. Introduction

For decades, three-dimensional (3D) city models have been primarily used for visualisation, with an increasing number of stakeholders and practitioners recognising their advantages in decision-making processes. A key reason for this is that the world around us, presented in 3D form, as opposed to two-dimensional (2D) maps and drawings, is typically more comprehensible and easier to perceive and can readily serve as a tool for communicating and sharing information [1].

Therefore, 3D city models play an essential role when analysing and managing urban data, and this is evidenced by the large number of applications that utilise this information [2]. The integration of 3D city models with non-geometrical data, such as social, economic, acoustic or historical information has proven its utility in fields including 3D cadastre [3], emergency response [4], decision-making, urban planning [5], smart cities [6] and more recently, in the digital twin field where 3D city models serve as a fundamental source of information [7].

Among the many elements that make up city models, one of the most prominent, if not the most important, are the buildings. Nevertheless, despite their usefulness, the publicly available 3D building models are often presented as coarse, solid blocks in most cases. Occasionally, these block models are augmented with simplified roof shapes and image texture [8]. However, details of the building openings (windows and doors) are often missing.

There is a significant body of literature focused on trying to automate building model reconstruction to make it low-cost, faster, accurate and easy to update. Nevertheless, many of the approaches proposed to solve the challenge of automatic generation of detailed models require a great deal of pre-and post-processing, therefore restricting their utilisation as standalone solutions [9].

This challenge is partly associated with the complex geometry of buildings, restricted access in performing measurements, or the presence of objects that obscure the façade, especially their lower parts [10]. An example illustrating the complexity of buildings based on their architectural style is presented in Figure 1. This diversity of styles poses problems to developing universal and robust methods for the reconstruction of building façades in the city environment.

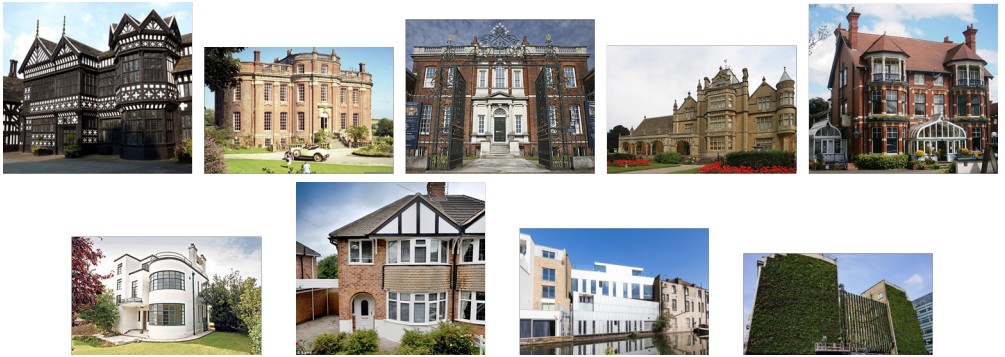

**Figure 1.** Examples of architectural styles in United Kingdom. Top row from left: Tudor, Baroque, Georgian, Victorian and Edwardian; bottom row from left: Art Deco, 1930s, Modern (white-grey building) and Green 'Living Wall'. The images were taken from multiple modalities : Modern: https://architizer.com/blog/inspiration/collections/architecture-on-the-market-london-contemporary-homes/, accessed on 17 January 2022, Living Wall: https://tugc.co.uk/portfolio-item/maintaining-one-of-the-biggest-living-walls-in-london/, accessed on 17 January 2022, Others: https://www.bohaglass.co.uk/british-architectural-styles/, accessed on 17 January 2022.

3D building reconstruction is relevant to a wide range of research fields, such as computer vision, computer graphics, photogrammetry, geodesy, architecture, civil engineering and construction. To avoid confusion, in this review, the concept of 3D building reconstruction encompasses the processing workflow consisting of several steps: data acquisition, data registration, scene interpretation and object extraction and 3D modelling (Figure 2).

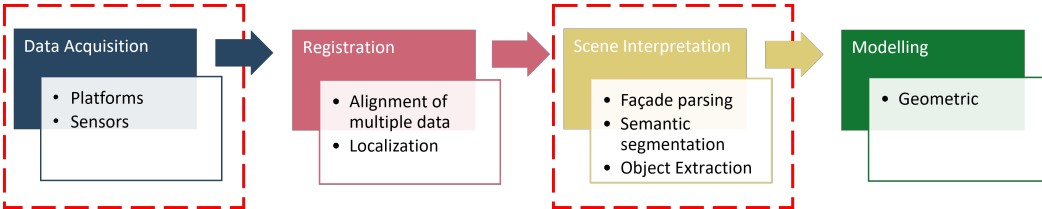

**Figure 2.** The general workflow of the reconstruction process includes data acquisition and registration, scene interpretation and modelling. Red boxes indicate elements of the process covered in this review for detailed façade reconstruction.

Data acquisition is a crucial step in building reconstruction. A building's architectural style, size, location (i.e., city centre versus suburbs) and the level of detail at which the

final product is to be presented can all determine the choice of the sensor technologies and platforms used to acquire the necessary data used as a basis for reconstructing a building. The next step in the 3D building reconstruction pipeline is data registration. In many cases, to obtain a complete picture of a building, it is necessary to combine data from different perspectives (e.g., terrestrial and airborne) and sensors. Multiple methods are used to register data from point clouds [11–16] or 2D images [17–21].

Scene interpretation is the next stage of the reconstruction pipeline. Many concepts relate to scene interpretation, including façade parsing, segmentation and object extraction. The primary purpose of these methods is to thoroughly extract and analyse the shape and position of walls and façade openings (windows, doors and balconies) from the acquired data. The final stage of the reconstruction process is the geometric modelling of the façade and its elements. These can be achieved using parametric [22], surface [23–25] or volumetric [26,27] modelling approaches.

To date, solutions for detecting the openings of the façade are scarce and generally limited in application to buildings with a relatively narrow architectural variety. This is due to, inter alia, a relatively monolithic type of publicly available data and from the generally small amount of available data that would enable more research to be conducted. Nevertheless, the number of applications that could take advantage of this knowledge of detailed 3D building models is increasing.

Therefore, this review initially focuses on highlighting the current research on data and key technologies used to enrich building façades, especially the methods used for façade parsing and building-opening detection. Accordingly, the registration of input data [28] and the geometric modelling of buildings [10,29] are beyond the scope of this review.

The first part of this review (Section 3) presents an overview of applications, such as building-information modelling and solar energy potential estimation, that could benefit from detailed 3D building models. Section 4 presents an overview of the platforms most commonly used to record building façades, in addition, we introduce emerging platforms, such as handheld scanners, the use of mobile phone images and volunteered geography information. Next, the advantages and disadvantages of range-based and image-based data used to identify façade openings are discussed (Section 5).

In Section 6, we discuss the methods developed to extract the façade elements from 3D and 2D data and highlight the challenges of the current approaches. Finally, we present the key limitations of the current methods and identify knowledge gaps in the current body of research, and we subsequently proposed key areas for future investigation.

## 2. Methodology

The approach taken to collate existing publications on detailed façade reconstruction for this review was based on an investigation of building reconstruction described in other literature reviews [10,30,31]. This enabled an initial understanding of what had been achieved within the building reconstruction field in the past and the identification of relevant keywords to search for related literature. To the best of our knowledge, there is no literature review dedicated to façade opening extraction, aside from that by Neuhausen et al. [32], which presented methods for window detection from images.

The increased technological development in data collection and associated processing methods has impacted the scientific community, as reflected in the increase in the number of journal articles, books and series on facade-related studies (Figure 3). The origins of research related to façade reconstruction involved the processing and analysing of images. About a decade later, in 2011, the possibilities of point clouds for building reconstruction began to be explored. The year 2016 represents the advent of research introducing deep-learning methods, which, in 2021, comprised about two-thirds of publications on the use of deep learning in façade opening extraction-related topics. To order the content of the collated literature into a manageable taxonomy, a set of themes and questions was developed to guide this review (Table 1).

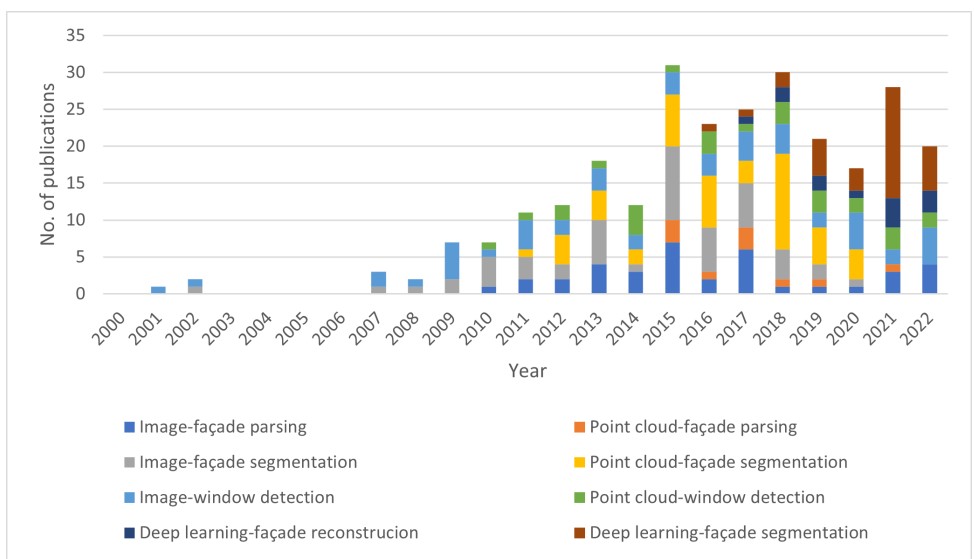

**Figure 3.** Number of façade opening extraction-related publications per year in journals, series and books according to the Web of Science portal based on the following keywords (in quotes) and Boolean Operators (in uppercases): (1) 'façade parsing' AND 'image', (2) 'façade parsing' AND 'point cloud', (3) 'façade segmentation' AND 'image' NOT 'indoor', (4) 'façade segmentation' AND 'point cloud' NOT 'indoor', (5) 'window detection' AND 'image' AND 'facade' NOT 'indoor', (6) 'window detection' AND 'point cloud' AND 'facade' NOT 'indoor', (7) 'façade reconstruction' AND 'deep learning', and (8) 'façade segmentation' AND ' deep learning'.

**Table 1.** Questions defining the main topics covered by the review.

| Review Section | Questions and Example |
|---|---|
| Section 3: Applications | What are the applications that could benefit from more detailed 3D building model? For example, solar potential estimation, building information modelling and energy analysis. |
| Section 4: Platforms | What measurement technology is used? What strategy can deliver reliable and suitable data for façade opening extraction? For example, airborne, ground level, mobile and static platforms. Are there any emerging technologies? Handheld, mobile phones and volunteered geography information. |
| Section 5: Data types \representation | What are the advantages and disadvantages of data used for façade opening extraction? What are the challenges of existing methods? For example, what characteristics of range-based and image-based data should be considered when extracting façade openings. |
| Section 6: Scene interpretations | How is the data processed? For example, analysis of point cloud density for façade opening extraction. Machine learning capabilities for façade opening extraction. What are the benefits of data fusion? |

## 3. Applications

Simplified and coarse representations of buildings can serve as an effective source for analysis on either a city or national scale. However, a lack of detailed information on the façade elements poses difficulties for analysis at street or building scale [33]. Despite much research, the reconstruction of buildings with semantically rich façades remains a challenge due to the complexity of the task or lack of adequate data. Nevertheless, the realisation of an increasing number of applications that could benefit from detailed information about

the composition and appearance of existing buildings has sparked growing interest among academic and commercial researchers (Figure 4).

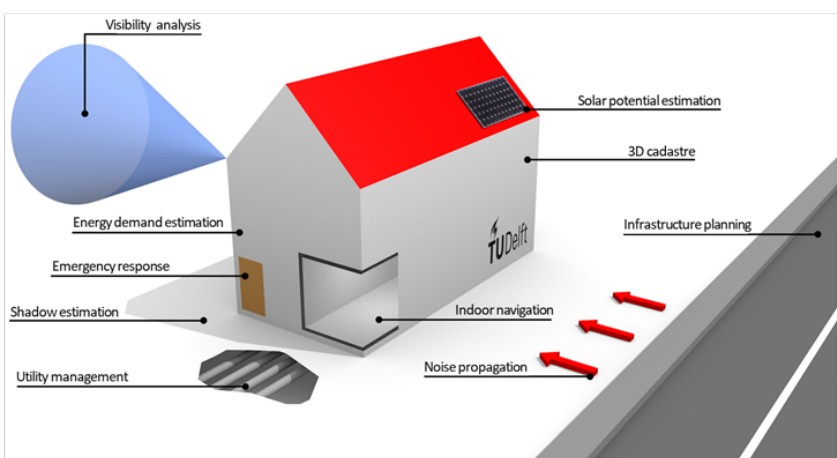

**Figure 4.** Applications of 3D building models for decision-making support. Reprinted with permission from Ref. [2].

### 3.1. Building Information Modelling

Among the applications requiring detailed façade information, building information modelling (BIM), solar potential analysis estimation and a broad spectrum of energy analyses are most common. BIMs are digital facility models containing rich semantic information. The potential of BIM is recognised among the Architecture, Construction, Engineering and Facility Management (ACE/FM) community, since BIMs facilitate communication between a range of stakeholders involved at different stages of a facility's life cycle.

BIM models, in most cases, are generated from blueprints at a level of detail that exceeds that which is typically obtained based on remotely sensed data [10]. However, despite the lower level of detail, creating 3D models of buildings directly from remotely sensed data for BIM is useful when blueprints are unavailable (e.g., historic buildings) or are old and outdated. In such cases, remote sensing methods allow the creation of a 3D representation of buildings in as-built/as-is conditions [34–37].

### 3.2. Solar Energy Potential

Solar energy potential estimation is another application for which detailed information about the façade and its elements is relevant. Roof surface analysis has been used for many years in solar energy estimation, often overlooking the possibilities of also exploiting walls. This is due to a tedious and time-consuming process if considering analyses on a city scale and not individual buildings [38] or the need for more sophisticated tools based on 3D geospatial data [39].

However, the growing interest in renewable energy policies has led to intensified research on the solar energy potential associated with vertical surfaces (e.g., walls) in the urban environment. This is of particular importance in the modern urban landscape because the area of a façade often exceeds that available on the roof. Another advantage of utilising vertical surfaces is that they are not typically affected by dust and snow cover during the winter months. In recent years, several studies have investigated the potential of solar energy from a building's façade (vertical surfaces) [39,40].

For instance, Desthieux et al. [39] assessed the solar energy potential for rooftops and façades in Geneva, Switzerland, by combining light detection and ranging (LiDAR) with 2D and 3D cadastre data (i.e., 3D building models, 2D roof layer, 2.5D digital surface model). Their experiment proved the practicality of 2D and 2.5D data for solar radiation analyses conducted on large urban scales. However, the obtained level of detail was insufficient when features, such as windows or balconies, were necessary for the analysis.

Catita et al. [40] investigated the relevance of roofs and façades for the photovoltaic potential of the University of Lisbon, Portugal, by combining aerial LiDAR data with 3D building models and solar radiation models. The authors presented a methodology addressing which façade is most suitable for solar panels or where on the façade a panel should be localised.

### 3.3. Energy Analysis

Detailed façade representations are also beneficial to applications under the broad spectrum of energy analysis. For instance, the size and orientation of windows can have a significant impact on the energy consumption of buildings [41]. To minimize energy wastage, the area of doors and windows should be less than or equal to 20% of the floor area [42]. Moreover, the number of floors (which can be estimated based on the number of windows) and the window-to-wall ratio, among other factors, can contribute to heat demand modelling [43].

Building height and the percentage of glaze were also among the many factors used by Nageler et al. [44] for dynamic urban building energy simulation. Other energy analysis applications that could make use of detailed façades include estimation of energy demand [45], energy-efficient retrofits planning [46], assessment of energy performance [47] and luminance mapping [48].

### 3.4. Civil Engineering

One of the important aspects considered by civil engineers is an assessment of the stiffness of buildings, the resistance of which can be weakened by ground movements (e.g., caused by underground construction or geohazards). In this case, information about building openings, and in particular windows, is considered since they affect the overall stiffness of the building structure [8,49,50]. Information on façade openings is also critical for evacuation planning and emergency response, where not only a geometrical representation of façade openings are essential, but also their semantic representations [4].

### 3.5. Other Applications

The applications highlighted above are domains within which the advantages of using more detailed building façades are clear. Other applications that could benefit from detailed façades include light pollution and shadow simulations [51], prerequisite for disaster management [52], movies and virtual reality [53], and potentially bird-window collision analysis [54–56] where the window angle of orientation plays an important role in assessing avian mortality. As outlined, the enrichment of volumetric 3D models with additional information about façade openings clearly has significant potential for opening new possibilities for more effective and accurate analyses and urban environment management.

## 4. Platforms

Although research into the reconstruction of 3D buildings has increased in recent decades (see Figure 3), the majority of studies have sought to improve the accuracy of determining the external geometry of buildings and marginalizing the detection and reconstruction of building openings. In general, the data used for the reconstruction of the façade can be grouped according to the viewing perspective from which they were obtained or the type of data collected. Regarding the perspective, aerial and ground-based platforms are mostly used due to their viewing angle. As for the sensor, the most commonly used data are those acquired using ranging-based and imaging-based sensors (Figure 5).

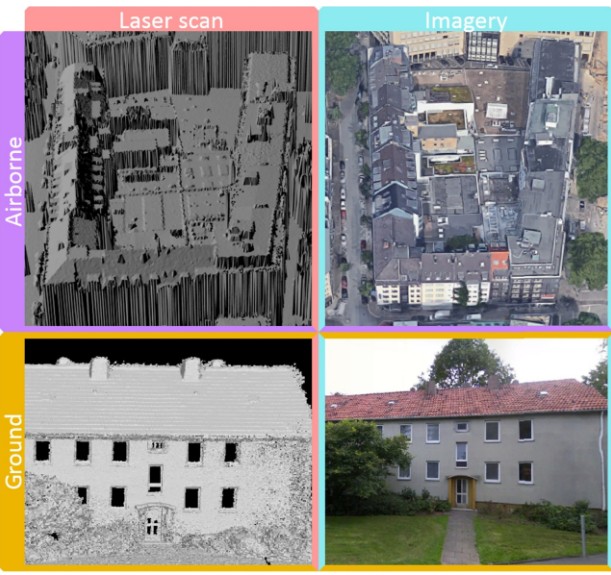

**Figure 5.** Data commonly used for 3D building reconstruction. Reprinted with permission from Ref. [50].

The main factors that determine the choice of the platform from which the data for 3D building reconstruction are acquired are the complexity and height of the specific building feature/component of interest (e.g., roof, façade). Figure 6 shows what information about the building can be collected depending on the level of the platform. For instance, data acquired from aerial platforms are most suitable for reconstruction of the roof [57] or the top of a tall building. In contrast, data collected from ground level captures parts of buildings that are not visible from a plan view, such as the façade and its openings.

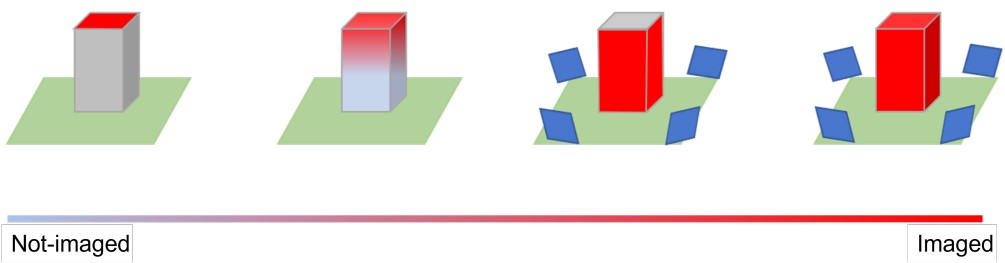

Not-imaged    Imaged

**Figure 6.** Building elements captured depending on the platform used. From left to right: nadir view allows rooftop detection, the oblique view extends nadir by mapping façades, terrestrial platforms capture façade information, and the combination of all allows mapping the whole building. Adapted with permission from Ref. [58].

Airborne platforms provide the most pertinent data for roof modelling, building footprint detection (oblique images), showcasing streets and mapping the locations of open spaces. However, the capabilities of airborne platforms for detailed 3D elevation modelling are limited. This is largely due to the downward-facing (nadir) viewing perspective offered by most sensors that are mounted to airborne platforms. Alternative solutions to mapping façades from airborne platforms include the use of sensors (e.g., cameras) that provide an oblique viewing perspective [59–61] or unmanned aerial vehicles [62].

An airborne survey allows information for a large area to be readily acquired in a single flight but at a high cost. A solution to this is through the use of unmanned aerial vehicles (UAVs). Nevertheless, the use of UAVs for surveying complex urban environments poses several challenges of its own. First, a UAV survey requires knowledge and appropriate certificates on how to operate the drones. Moreover, the area and height at which UAVs

can operate in cities are limited due to the safety and security risks posed by skyscrapers and overhead power lines.

In addition to airborne platforms, data obtained from ground-based platforms are used to gather information for the 3D reconstruction of buildings. This group of platforms includes both static and mobile units (e.g., vehicle, backpack, robot). Static ground platforms consist of a sensor most often placed on a tripod, with data collected in stop-and-go mode. This means that, once all the required data are collected from one measuring point, the equipment is then moved to the next point to perform measurements from a different perspective [63]. Datasets acquired using static scanners are typically large, therefore, requiring considerable storage and time for subsequent processing. Moreover, collecting data for multiple buildings is time-consuming.

The availability of Global Positioning System (GPS) technology contributed to the creation of mobile mapping systems (MMS) [64] in which sensors are mounted on moving platforms (e.g., vehicle, rail cars, robot and backpack). Most commonly, mobile technologies are equipped with sensors, such as a scanner and camera, Global Navigation Satellite System (GNSS) and Inertial Measurement Unit (IMU), with data collected along the driving path. Mobile systems reduce the time and cost needed to undertake a survey, as they can be used to acquire data from large areas in a shorter amount of time.

In recent years, manual simultaneous localization and mapping (SLAM) based laser scanners have emerged, allowing the relatively seamless collection of 3D data (Figure 7). These handheld scanners are growing in popularity due to their measurement speed, low cost and ease of use. These devices combine different sensors (LiDAR sensors, RGB camera and IMUs), the data of which are used by SLAM technologies to track the position of the device while creating a mapped environment [65]. Handheld scanners are especially useful in the absence of a GPS signal which makes it difficult to determine the position of the sensor or platform (e.g., satellite signal blockage due to building, bridges, trees, indoor surveys and 'multipath' effects). Solutions based on the SLAM approach allow for faster and cheaper data acquisition at the expense of lower accuracy compared to traditional methods.

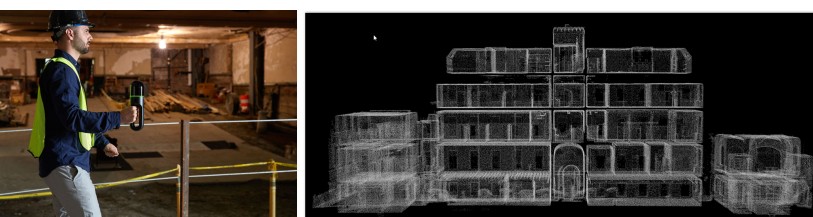

**Figure 7.** An example of handheld technology. Source: Leica.

Over the past decade, in addition to traditional solutions, there has been a growing interest in the geospatial community to use images acquired from smartphones. This data collection method is cheap and fast and enables the acquisition of images from different perspectives, making it possible to register façade elements not visible from other platforms. An additional advantage of mobile phones is the GNSS sensor with which the device is equipped.

The GNSS sensor allows for a relatively accurate determination of the position of the camera when taking an image, depending on the quality of the signal and the sensor [66]. The potential use of images acquired from mobile phones for building modelling has been investigated in several studies [67–69]. In these, phone images were used to create point clouds of buildings through the structure from motion (SfM) [70] algorithm. The presented results demonstrated that data acquired from mobile phones by multiple users could be successfully combined to derive a 3D point cloud [67] (Figure 8).

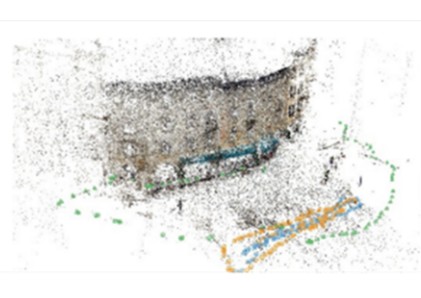
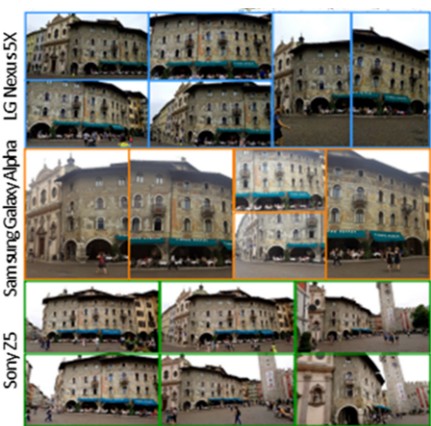

**Figure 8.** 3D building reconstruction from mobile phone images. Image boundary associate with the colour coded cameras in the point cloud. Reprinted with permission from Ref. [67].

Data collection from multiple users echoes the concept of 'citizens as sensors' of Goodchild [71], which opened up the world of 'volunteered geography information' (VGI) and 'volunteered geography' (VG) [72]. Due to the adoption of crowd-sourcing, many non-experts can now share geospatial data. For example, Fan et al. [73] designed an integrated platform to reconstruct 3D building models based on images collected by volunteers or non-expert users. The results appear promising for delivering more detailed building models, however, no information on the location accuracy was reported.

Despite the undisputed positives associated with citizens actively participating in data collection, this approach raises questions about the quality assurance [74]. As for using crowd-sourced data for a more detailed reconstruction of buildings, a set of well-designed data acquisition guidelines could ultimately help enhance façade openings detection.

For example, many images from different locations, collected under various conditions, could allow for more accurate recognition of city elements and thus the creation of more robust and flexible methods for openings detection and modelling. In addition, such data could be used to develop databases for training, validation and testing new strategies, where dedicated and diverse datasets could assist data-driven approaches.

Data acquired from terrestrial platforms enable the collection of information on the lower parts of buildings, which allows subsequent identification of building openings. Nevertheless, data collected from this level poses several limitations. First, permission to enter private properties (e.g., residential neighbourhoods) may be restricted for legal reasons. Furthermore, objects on the ground can be obstacles that limit a sensor's field of view. These include, among others, pedestrians, passing and standing cars or street furniture. These limitations pose difficulties in obtaining a complete dataset for a given area of interest and, consequently, lead to problems with recreating a complete and detailed 3D building model. Missing façade elements can be supplemented with data from aerial platforms obtained with an oblique camera. Although reconstructing detailed building façades is possible, it is still a challenging task. This is mainly due to constraints associated with the complexity of a building façade, occlusion and camera angle, radiometric changes in illumination and perspective in different images, sensitivity to sun glint and hot spots [75,76]. This leads to incomplete coverage and holes in the data that make it difficult to correctly identify objects or problems with image matching that can be used to replicate the point cloud from images (Figure 9).

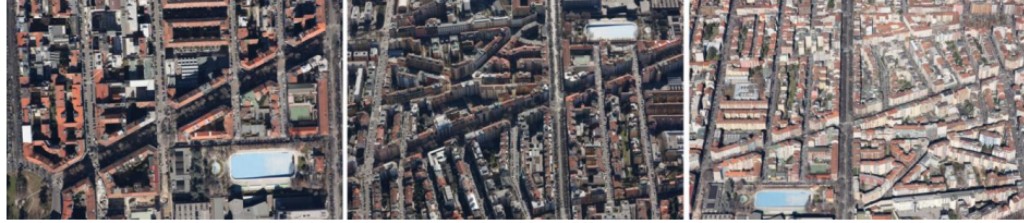

**Figure 9.** Left: nadir, middle and right: oblique views of the same urban scene. The differences in radiometric resolution between the views, cause difficulties in image matching. Reprinted with permission from Ref. [77].

## 5. Data Types/Representation

The two most popular measurement techniques used in building reconstruction are laser scanning and photogrammetry. The basics for these techniques can be found in McManamon [78] and Mikhail [79] and Wolf et al. [80], respectively. Point cloud datasets obtained from laser scanning or multi-view stereo vision are commonly used for 3D building reconstruction and city mapping (Figure 10). Compared to data, such as 1D measured distance or projected 2D images, point clouds provide high-resolution data for accurately determining the shape, size and position of an object in 3D space, enabling object reconstruction and modelling [81].

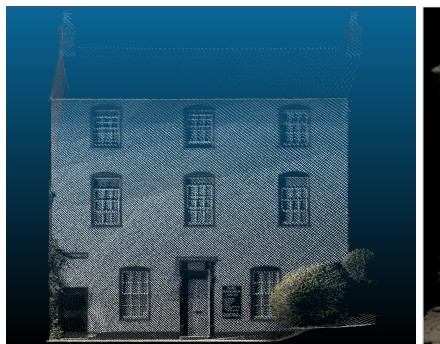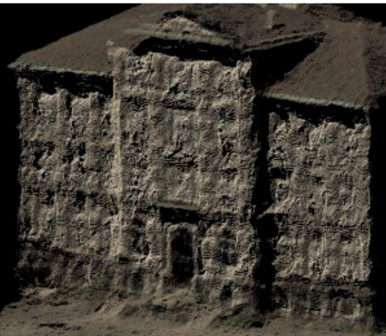

**Figure 10.** An example of point cloud from terrestrial laser scanning (left, source OS © Crown copyright and database rights 2022 OS and dense multi-stereo matching from oblique aerial imagery (right, reprinted with permission from Ref. [82]).

Range-based point clouds are quick to obtain, not affected by illumination and of high accuracy, which makes them a good source of data for 3D building reconstruction. However, the range-based point clouds have shortcomings which include lack of semantic information, high costs of the equipment and high technical requirements for operation [83].

Additionally, the distribution and density of points obtained is not always uniform and depends on the distance between the sensor and the object. A subsequent lack of information, high redundancy and low distribution at the edges of objects are all factors that can lead to difficulties in the reconstruction of complex buildings [84,85]. Moreover, differences in the accuracy and density of point clouds do not allow for a simple fusion of data obtained from different viewpoints and sensors [86].

Image-based point clouds are generated by applying triangulation to stereo-image pairs, such as through SfM. One of the main advantages of image-based point clouds is that they contain high-resolution textural and colour information, which enables the generation of photorealistic models [87]. Nevertheless, point clouds obtained from SfM can suffer from non-uniformities and may contain a higher noise level than range-based point clouds. Compared to the raw point cloud from range-based methods, in which the 3D information is obtained directly from the implicit scale factor, image-based point clouds require this information to be extracted from the images using camera parameters (i.e., intrinsic and extrinsic parameters).

One of the essential differences between the two types of point clouds is the accuracy, which, for range-based point clouds, is greater in the depth direction compared to image-based point clouds. On the other hand, the accuracy of the image-based product is greater in the direction perpendicular to the depth plane.

One of the point cloud characteristics used in detecting building openings is the presence of holes (i.e., data gaps) where glass objects are located, which is due to signal penetrating through the transparent surface (range-based) or difficulties with image matching for glass/transparent surfaces (image-based). This feature is helpful in the detection of elements, such as windows or glass doors; however, when the objects are covered (e.g., curtains and blinds), their detection based on the missing data is challenging. Figure 11 illustrates a point cloud with various examples of windows. As presented, the assumption that a hole represents windows in a point cloud can be misleading. Specifically, some of the windows are partially or entirely obscured, meaning that the holes in the point clouds do not match the actual shape of the window.

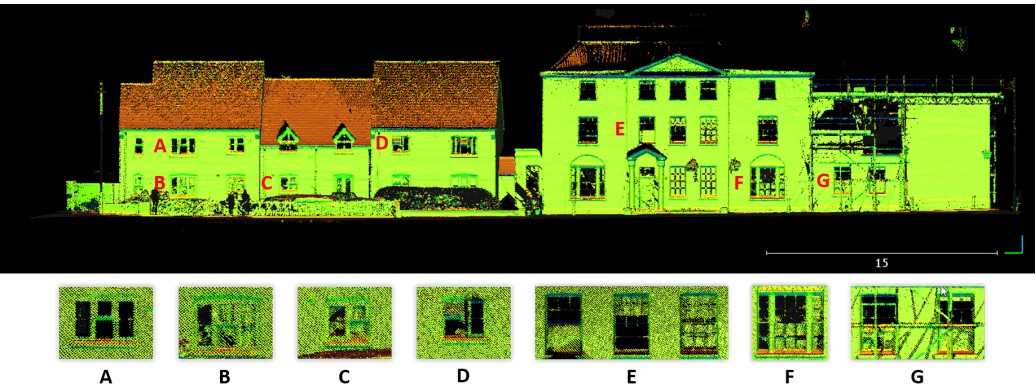

**Figure 11.** Mobile mapping system point cloud representing a building façade. (**A**)—window represented by holes without noise. (**B**–**G**)—samples of windows with varying degrees of obscuring. Source: OS © Crown copyright and database rights 2022 OS.

In addition to using images to create a point cloud, 2D image data itself provides a source of information that can be used for the semantic enrichment of building façades. For instance, the edges of objects are easier to detect in images, which allows the boundaries of objects to be easily defined. Furthermore, the colour information (presented in different colour spaces) can be used to segment the façade. Recky et al. [88] successfully adapted a k-means clustering in a CIE-Lab colour space to detect windows on complex historical buildings. Despite its advantages, the reconstruction of buildings based only on photogrammetric methods provides several challenges. Occlusions related to vegetation, pedestrians or vehicles, duplicate measurements of the same object, or insufficient data volume related to site accessibility are among the main factors that prevent an accurate reconstruction of objects.

## 6. Scene Interpretation

During the past decade, a large body of research has been conducted on 3D building reconstruction. Nevertheless, most of this deals with the manifold facets of the buildings. Specifically, there has been a primary focus on improving the accurate representation of the external shape of the building without consideration of its details. To the best of our knowledge, the work by Musialski et al. [30] is the most comprehensive review of the methods used for automatic and semi-automatic reconstruction of buildings from images and LiDAR data, which includes methods for window detection from image data.

The number of studies that addressed the subject of façade opening detection prior to 2013, when Musialski's work was published, was minimal. However, recent advances in

data processing approaches (e.g., computer vision and deep learning) and more accessible hardware has contributed to the growing interest in detecting façade openings.

Here, we present an update of more recent research on building reconstruction, with a focus on façade elements identification. With respect to scene interpretation methods, we assume that segmentation refers to the assignment of a label only to known objects, whereas scene parsing involves assigning annotations to all points. In contrast, object detection selects specific elements (e.g., windows and doors) from a given dataset.

### 6.1. 3D Scene Interpretation

Essentially, point clouds generated from laser scanning or multi-view stereo vision are commonly used for 3D reconstruction of buildings due to their ability to significantly enhance geometry reconstruction and surface modelling [81]. Interpreting scenes obtained from point clouds to extract information about the elevation of a building and its openings can be divided into two stages. The process begins with segmentation that involves grouping points belonging to one surface (e.g., a wall). These regions are further subdivided to obtain more detailed façade information (e.g., doors, windows and balconies) using feature extraction methods.

### 6.1.1. Segmentation

Although the segmentation process does not directly translate into the delineation of the façade openings, it is a process that enables the distinction of those points within the point cloud that belong to the building (Figure 12). The general concept is to group points into homogeneous components based on the common features to exclude elements that are outside the area of interest, thus reducing the number of points and allowing for quicker processing. The advantages and disadvantages of common point cloud segmentation methods are presented in Table 2 and further information can be found in the work of Wang et al. [89] and Xu and Stilla [10].

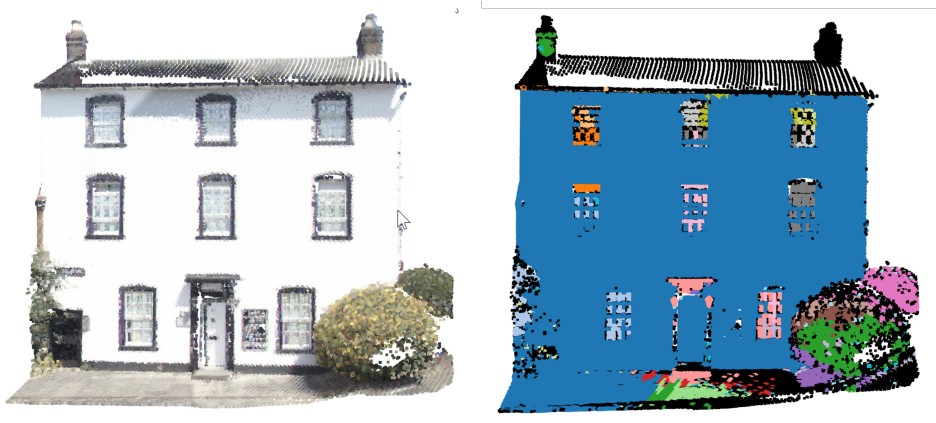

**Figure 12.** Example of building segmentation. Raw point cloud collected from terrestrial laser scanner with RGB values. Source OS © Crown copyright and database rights 2022.

**Table 2.** Summary of different point cloud segmentation methods for building reconstruction

| Method | Advantages | Disadvantages | Examples |
|---|---|---|---|
| Model-based | Resilience to outliers and noise | Computationally expensive, time-consuming, high memory consumption, difficulties with reconstruction of objects for which there are no mathematical expressions | [90] |
| Region-growing | Good preservation of edges and boundaries of surfaces and artifacts | Sensitive to outliers and noise, over segmentation | [91–93] |
| Clustering-based | Easy to implement, no need for setting of seeds | Over and under segmentation, high computational costs, points on the edges may meet the requirements of more than one cluster | [94,95] |

### 6.1.2. Scene Parsing

Façade parsing is a crucial step in the 3D building reconstruction process that leads to the classification of input data into semantic regions, such as walls, doors and windows [96] (Figure 13). Façade parsing enables a better urban scene understanding and enables more effective information storage on building façades. However, most examples of point cloud semantic segmentation of the urban environment do not include the detection of openings in the building façade.

Li et al. [24] introduced a façade parsing approach for which a point cloud is first decomposed into depth planes. Next, façade elements are detected using a combination of machine-learning-based classification and prior semantic knowledge. The overall performance of the method presents good results. However, it can be challenging to apply if the occluded area varies from the visible façade elements, hindering accurate reconstruction of the occluded parts of façade. To improve the method, the authors suggested considering the use of weak architectural principles or fusing the point cloud with images or GIS data.

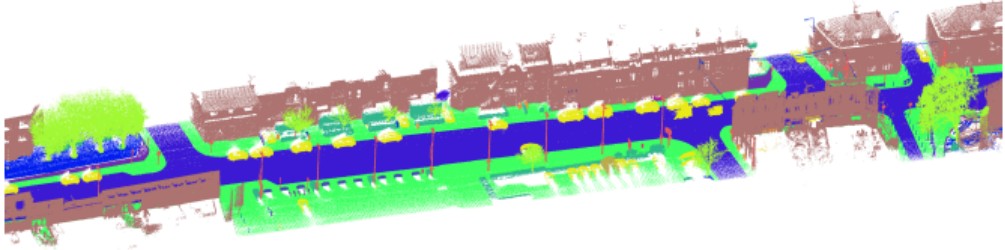

**Figure 13.** Part of Paris-Lille-3D dataset. Semantic labels for each object class, with unique colour for each class. Reprinted with permission from Ref. [97].

### 6.1.3. Façade Openings Extraction

The next step in the detailed 3D façade reconstruction is the extraction of the façade openings. Current façade openings detection strategies for point clouds can be categorized into hole-based and rule/symmetry-based (Table 3). Most building openings have glass

elements, and their characteristic 'transparency', which generates information gaps in the point cloud, can be exploited for the detection of openings [98,99]. A second characteristic often used to detect openings from point clouds is their typical rectangular shape represented by two vertical and horizontal edges.

Hole-based opening detection methods search for empty spaces in a point cloud and rely heavily on the point cloud density. The reason for this is twofold. On the one hand, a dense raw point cloud for walls and low point density for glass objects allows the detection of gaps in the point cloud. On the other hand, a low raw point density may be insufficient for the reliable generation of feature geometries [100].

Many studies on hole-based opening detection use terrestrial laser scanners (TLS) since their data density typically exceeds that of point clouds obtained from mobile mapping systems (MMS) or aerial laser systems (ALS). Pu and Vosselman [101] tested their method based on a Triangular Irregular Network of data from TLS (average point density 500 pts/m$^2$) and MMS (around 100 pts/m$^2$); while the method recovered the outlines of building openings from the TLS data, the point cloud density from MMS was too low to detect windows and doors.

High-density point clouds of the urban environment are not always available as TLS data collection, compared to MMS and ALS, is more costly and time-consuming when surveying large areas. The additional inconvenience of a highly-dense point cloud is the additional data storage and management required.

To address these issues and assess the effect of point cloud density on correct façade opening detection, Zolanvari et al. [102] tested their slicing method on three buildings of diverse size and complexity. They suggested that a density of 130 pts/m$^2$ should be sufficient to achieve at least 90% of accuracy. Although hole-based methods allow for façade opening detection, they present several shortcomings, such as sensitivity to outliers and noisy points, user intervention [101,103,104] and challenges with detecting windows covered by curtains [104–106].

Another façade opening detection strategy for 3D point clouds involves rule and symmetry-based methods. Rules play an important role in detailed façade reconstruction as openings follow architectural rules concerning their shape and spatial relationships (e.g., the locations of windows and doors relative to walls). Rules can be applied to openings extracted from corner detectors [9,107], a graph model [73], machine learning [108] or based on the assumption of symmetry in window distribution [109].

Applying rule and symmetry-based methods allow detection of openings from partially missing data. However, these methods are heavily depend on symmetry, and repetitive elements can affect their robustness [110]. Moreover, the approach is unsuitable for low-rise buildings, as vertically repeated features are unlikely to be present. This diversity, complexity and asymmetry of the façade structure does not allow for the creation of flexible rules describing the façades and their elements, posing problems in automating the reconstruction process of detailed 3D building models.

To address this problem, Fan et al. [73] proposed façade layout graph model method based on Gestalt [111] principles and the principle of architectural form [112]. Although the proposed method describes different types of façade structures, its applicability for complex structures and automation needs further improvements.

The methods outlined are often tested on a limited number of datasets that are lacking in the representation of buildings with diverse architectural styles. The lack of benchmark datasets with various styles makes it difficult to develop and test new methods (e.g., deep-learning based) for façade opening detection or point cloud parsing. Despite a recent increase in the number and availability point clouds of urban environments, their classification does not include façade openings.

**Table 3.** Summary of different methods for façade opening extraction from point cloud.

| Method | Advantages | Disadvantages | Examples |
| --- | --- | --- | --- |
| Hole-based | The assumption that windows are an integral part of the façade reduces the search area | Depends on point cloud density, misclassification due to occlusion holes, points reflected from curtains may be similar to those reflected from wall causing difficulties in distinguishing façade openings, high-density point cloud require more data storage | [100–106,113,114] |
| Rule-symmetry-based | Successful with low resolution data, allows detection from partially missing data | Necessary user intervention to create rules, high reliability on symmetry and rules, difficulties with application for non-symmetrical façade types | [9,73,107,108,115] |

### 6.2. 2D Scene Interpretation

While this review focuses on 3D building reconstruction, it is also crucial to outline methods used to detect façade elements from 2D data for two main reasons. First, there is increasing amount of data being collected by MMS that, in addition to the acquiring images, also provide the camera model parameters, GPS and IMU data [116], allowing for 2D to 3D projection of data. Secondly, the 2D images allow the isolation of features with linear components (e.g., windows and doors) that can improve the quality of the final detailed building model [117,118].

#### 6.2.1. Scene Parsing

Image semantic segmentation is a process of adding a semantic class to every image pixel. Generally, segmentation-based methods for extracting façade elements from 2D images can be divided into two groups (Table 4). These are: (i) bottom-up methods [119–123]; (ii) grammar driven (top-down) methods [124–126].

Bottom-up methods solve the image segmentation problem at the pixel level. Although bottom-up methods deliver satisfactory pixel-wise results, a result of the segmentation is not suitable for every application due to divergence from basic architectural rules; for instance, irregular shape (pixel-wise segmentation may not result in straight and even lines) and incorrect size of windows or doors. These irregularities may be enough to estimate the glass-to-wall ratio, which might be suitable for thermal related applications.

However, their usability for constructions-related applications, such as BIM, is insufficient. The reason for pixel labelling challenges is that radiometric variation in images, dynamic or static occlusion, and shadows can hinder the precise recognition of objects. The above-mentioned issues can lead to a partial or complete gap in the segmented scene. To improve the quality of bottom-up methods, weak architectural constraints can be applied [127–129], thus, reducing occlusion issues and improving the overall accuracy; nevertheless, issues, such as the window's vertical alignment, still persist.

Recent bottom-up methods rely on convolutional networks, however, the lack of datasets with diverse architectural styles poses difficulties for 2D data segmentation [130,131]. Specifically, Lotte et al. [130] trained a neural network on different available datasets and then

tested it on an entirely new dataset. The results suggest the need for accurate annotation of the images and the lack of an exact extraction of façade elements on an unknown data set, except where training and testing datasets share similar features. Moreover, Kong and Fan [131] note that existing benchmark datasets constitute rectified images without distortion, which are different from those acquired from the mobile mapping systems. Hence, at present, methods developed on specific datasets cannot be easily transferable to other scenarios.

Top-down methods segment images based on grammar rules. As opposed to bottom-up methods, the implementation of grammar rules supports the prediction of the façade element's location, which is particularly important in the case of occlusion or changes in illumination. Moreover, applying the grammar rules restrict the area in which the algorithm searches for elements, as it considers only these segments that follow the rules.

Nevertheless, grammatical rules involve significant human intervention and prior knowledge to construct the rules describing a given architectural style, making it difficult to automate and adapt the methods to different data or sites. To reduce user intervention automatic learning of rules with ground-truth image annotations have been proposed [124,132].

For example, Gadde et al. [124] experimented with applying a set of grammar rules learned on one architectural style to another style. Their results indicated that the location of the façade elements is not ground truthful, therefore demonstrating the problems with the interchangeable application of grammatical rules.

**Table 4.** Summary of different methods for façade parsing from images.

| Method | Advantages | Disadvantages | Examples |
|--------|-----------|---------------|----------|
| Bottom-up | Better performance compared to rule-based methods [131], less prior-knowledge necessary | Susceptible to changes in lighting, irregular shape of the output (e.d., edges are not straight lines), occlusion hinders precise segmentation, lack of diverse training datasets | [53,100–102,104,106,113,114, 119,130,131,133–135] |
| Top-down | Higher prediction of façade element's location, better structure of detected façade elements | Rely on strong prior knowledge, difficulties in applying rules for one architectural style to other, lack of flexibility when facade is changed | [124–126,132,136] |

6.2.2. Façade Openings Extraction

The studies presented above focus on parsing an entire building façade, however, there is a considerable body of literature dedicated to window detection from 2D images. Neuheusen et al. [32] published the most recent overview of window detection from images. In that review, the methods are categorized into the following groups: grammar-based [137], image processing [138] and machine learning [122]. The same author has since focused on window detection from image data through a series of studies on window detection for risk assessment analysis [8,50,139].

These specific studies involved the assessment of a soft cascade classifier for window detection. Through a series of experiments, the authors proposed a final three-step pipeline. The process commences with image rectification, then the detection of objects is performed by a sliding window detector with a soft cascade classifier consisting of thresholded Haar-like features (Figure 14). The last step concerns post-processing to refine the detection step.

In future work, the authors plan to categorize windows by a type similar to that presented by Lee and Nevatia [140].



**Figure 14.** Example results of windows (presented as green rectangles) detected in image. Reprinted with permission from Ref. [50].

One of the main elements of the façades is entrance doors, the location of which could find applications in navigation [141], emergency response and flood analysis. Nevertheless, there has been little research dedicated to external front door detection. In contrast, there is a considerable body of research on door detection and localization in indoor environments. This has largely been driven by a growing demand for a variety of applications, such as navigational assistance [142] and 3D building reconstruction [143].

Many of these case studies are associated with mobility restrictions for disabled people or intelligence systems allowing safe navigation through indoor spaces. Although, in theory, the appearance of doors from inside and outside is identical, their characteristics differ significantly in terms of their surroundings, occlusion and lighting, which limits the interchangeability of the same methods for door detection in varying environments.

In the case of entrance door detection in outdoor scenes, Liu et al. [144], Kang et al. [145] and Talebi et al. [146] tackled this by implementing vision-based techniques, such as edge detection, line segments or local features to detect front doors from RGB images. The limitations of these studies are that they use small and relatively simple datasets to assess the results [147] or, in the case of Talebi et al. [146], have challenges with door detection from strongly tilted images and when the door and wall are almost identical in colour.

*6.3. Data Fusion for Scene Interpretation*

Many approaches to reconstructing façades utilise single-source datasets, where data from one source (e.g., camera and scanner) but multiple viewpoints are linked together. While this approach can cover a large area of interest, it does not mitigate or eliminate the limitations associated with a given technology. For example, point cloud data are unstructured, and the features determined directly from them are characterized by low geometric continuity. On the other hand, 2D images, which can be registered for real-world scale according to a point cloud, make it possible to describe the spatial distribution of elements in the image (e.g., linear features).

Therefore, the combination of the features extracted from both images and a point cloud allows for a more accurate reconstruction of building façades and their elements. Such a solution can be synergistic through supplementing the limitations of one method with the advantages of another (Figure 15), providing more reliable information [148].

| | Camera | LiDAR | Camera + LiDAR |
|---|:---:|:---:|:---:|
| Object detection | 🟡 | 🟢 | 🟢 |
| Object classification | 🟢 | 🟡 | 🟢 |
| Distance measurement | 🟡 | 🟢 | 🟢 |
| Object edge precision | 🟢 | 🟢 | 🟢 |
| Range | 🟢 | 🟡 | 🟢 |
| Colour | 🟢 | 🔴 | 🟢 |
| 3D reconstruction | 🔴 | 🟢 | 🟢 |
| Contrast, resolution | 🟢 | 🔴 | 🟢 |
| Poor weather performance | 🟡 | 🟡 | 🟡 |
| Low-light performance | 🟡 | 🟢 | 🟢 |

**Figure 15.** Sensing comparison: LiDAR vs. Camera. Green refers to good capability, yellow to fair and red to poor capability.

Although LiDAR and cameras are the most popular data sources used for 3D building reconstruction, both methods have limitations, that can be overcome by fusing data of different modalities. Accordingly, multimodal sensor fusion can increase the level of detail and quality of a building model and enable the creation of more information-rich 3D city models. For instance, incorporating spectral information from hyperspectral imagery would allow identification of the material properties of a wall.

Furthermore, combining data from thermal imagery allows for thermal attribute mapping [149–151], while multispectral information can help assess the conservation state of a building [152]. An interesting example of multimodal data fusion is reported by Jarząbek-Rychard et al. [153]. In that study, 3D scene reconstruction is obtained from Red-Green-Blue (RGB) and thermal infrared (TIR) images. The authors investigated the applicability of spectral and geometric features for thermal analysis and the usability of a 3D thermal point cloud for façade opening extraction through a set of experiments.

The same type of data was used in Lin et al. [154] to detect windows through a combination of geometric features and thermal attributes. One of the fundamental assumptions of this approach was that there is a significant difference in temperature between windows and their surroundings (e.g., a wall). In some cases, due to the reflections from warm objects, this temperature difference was apparent and, therefore, caused misclassification. Additional grammar rules and topology analysis could help improve the performance of this method.

The fusion of 3D point clouds and 2D optical images for façade feature extraction has been the subject of several studies [98,118,148,155]. Pu and Vosselman [118] used point cloud data to establish the general structure of a building façade before applying computer vision edge detection on the images. Linear features in the images were then compared with model edges in order to improve the final result. The authors concluded that 3D point clouds enable straightforward extraction of areas and surface normals, which is challenging with 2D image data.

On the other hand, image data outperforms point clouds when detecting boundaries due to overall higher image resolution. Wang et al. [155] explored the fusion of a 3D point cloud and image data in three steps. First, the structural information about a building façade was retrieved from images. This was followed by the exploration of different methods for mapping between the image feature and the 3D point cloud. In the last step, the point cloud optimisation is performed by considering the structural information.

The potential limitation of the method is the difficulty with the feature extraction of complex façades and missing data due to occlusions. Becker and Haala [98] integrated information from a TLS and camera with a 3D city model. In the first stage of this approach, point cloud data was used to retrieve larger building parts, and then the images were processed to extract more detailed elements, such as window crossbars.

## 7. Limitations and Challenges

Musialski et al. [30] identified three challenges affecting the methods of that time: full automation, quality and scalability and acquisition constraints. Despite the passage of time and the development of new algorithms and more sophisticated hardware, these problems still persist nearly a decade later. Moreover, a few more factors can be added to the list of limitations highlighted by Musialski et al. [30].

- *Automation:* Automation of 3D building reconstruction has been an aim for researchers over the last couple of decades. However, the complexity of this task is high since we seek to reconstruct a well-defined underlying object; however, as we do not know the underlying object, it needs to be estimated from the data. Moreover, combining the above-mentioned data acquisition methods with the presented data processing techniques still relies upon many manual steps. Many new approaches require human intervention in setting up processing parameters, resulting in low levels of automation. Improving automation of the 3D building reconstruction by limiting the human input should involve designing concepts that create more intelligent and flexible workflows.

- *Quality:* The reconstruction of 3D buildings is directly related to level of detail (LoD) and level of accuracy (LoA), which are selected according to the application of the 3D model, method of data acquisition, processing and labour costs. The current low level of automation in the 3D reconstruction of buildings leads to the necessity of operator intervention. This intervention may concern data capture, data quality control, data processing settings, or the final result of the reconstruction assessment.
  Each operator has different knowledge and experience, which may affect the quality of the final 3D model. Therefore, it is imperative to develop good practice guidelines to eliminate the subjectivity between operators. Automation can be the solution to this problem. Hence, it is necessary to develop solutions for autonomous data processing that would minimize the need for intervention by an experienced operator.

- *Data acquisition:* Due to its complexity, it is challenging to obtain a complete dataset for an urban environment. For instance, differences in the dimensions of the buildings lead to differences in the characteristics of the acquired dataset, e.g., a terrestrial point cloud captured for a tall building will be less dense in the higher parts as the sensor's distance from the target will increase. Moreover, issues accessing buildings can result in gaps in the acquired point clouds and hence lower quality of the data used in 3D building reconstruction. Incomplete data are often the result of static objects (e.g., vegetation) or dynamic objects (e.g., vehicles and pedestrians) obscuring parts of buildings, while data acquisition at different times of the day may help limit dynamic occlusions [156], removing static occlusions is a more complex problem.
  In this regard, the fusion of data collected from various platforms offering different fields-of-view can help reduce the impact of static occlusions. For example, data from oblique cameras show blueprints of the building, which are hidden behind the stationary occluding object in the ground-level image. Another factor limiting data acquisition is the sensitivity of optical data to weather conditions (e.g., shadow and direct sunshine), resulting in a non-uniform image. One solution to this problem is incorporating data from laser scanning, which is more resistant to lightning and weather conditions.

## 8. Knowledge Gaps and Future Directions

- *Deep-learning implementation:* There been a recent rapid increase in the development and use of deep-learning algorithms for processing point clouds and images. However, the full application of these methods in reconstructing 3D buildings is still in its infancy. One of the reasons for this is the lack of proven, state-of-the-art methods that could be practically applied in commercial projects. In addition, most of the developed algorithms focus on modelling indoor environments that differ from the outdoor environment characterized by lower quality data (higher noise and outliers).

The significant differences between outdoor and indoor environments therefore preclude the opportunity to interchange techniques used to reconstruct objects within each of these environments. Additionally, deep-learning approaches require a significant amount of test data to train the algorithm in order to recognize individual elements. The preparation of such data requires considerable time and manual work. Data collection itself is an expensive task, especially in the case of laser scanning over vast areas, and this is partly responsible for limiting the number of available datasets that could be used for training an algorithm.

A possible solution to overcoming the requirement of an extensive training dataset is the current data-centric deep-learning trend that emphasizes data quality rather than quantity. Focusing on well-prepared training data instead of the selection of the hyperparameters of a deep-learning model could help in the development of more robust workflows for the reconstruction of 3D building models.

- *Benchmark Datasets:* Publicly available benchmark datasets are essential for the development and evaluation of new methods and algorithms since they enable the direct comparison with the existing strategies [157]. Despite the advantages and demand for benchmark datasets, the number of suitable, readily-available datasets available for the detailed segmentation of building façades remains low. One of the reasons for this may be the relatively recent interest in building modelling beyond its traditional representation in the form of a volumetric model.

  Additionally, let us assume that the façade openings are not only to be located on an image or a point cloud but also modelled (i.e., fitted into the body of the building). In this case, information is needed to reconstruct the object's location in space, while it should not be an issue with point clouds, 2D image-based benchmark datasets need to include ancillary information, such as camera calibration parameters and sensor locations.

  Only such sets of comparative data permit an accurate validation of the results of 3D building reconstruction approaches. An additional requirement for creating a versatile and universal benchmark dataset is sufficient diversity of feature classes and their corresponding labels. Although currently low, the number of available benchmark datasets for complex urban environments is increasing with the advent of smart cars and autonomous vehicle applications (Table 5).

  Even though these datasets, which often consist of both 3D and 2D data, have a specific category of 'building', they do not typically include sub-categories, such as windows or doors. This is, however, slowly changing and since 2020 there has been a growing interest in point cloud data that includes façade-level classes, such as windows, doors, balconies [158,159]. Benchmark datasets based only on images contain more detailed classes, which sometimes include windows and doors, however, they often lack the internal and external parameters, therefore making 3D reconstruction of building façades challenging.

  Table 5 summarises benchmark datasets commonly used for detailed façade segmentation (with camera images) as well as data from multimodal sensors with the potential to be used for complex 3D building reconstruction. However, it is worth recognising the problems related to the use of benchmark datasets presented in Table 5, especially the fact that these datasets were not explicitly designed for the detailed reconstruction of buildings. Benchmark datasets are designed with a specific application in mind; however, in many cases, their use goes beyond the boundaries of the tasks for which these datasets were originally created [160].

  In a recent study by Koch et al. [161], the problem of 'heavy borrowing' of datasets within the machine-learning community was noted, where data created to solve one problem are used to solve another task, which can lead to misalignment. Many of the available benchmark datasets are generated in a specific geographic location, and using them elsewhere may yield poor outcomes. For example, a deep-learning

model trained on data collected in the United States may not perform well on images from Asia.

Additionally, factors, such as the sun path, weather conditions, variety of backgrounds and different camera specifications, can affect the model's accuracy. At the moment, there are no available well-constructed benchmark datasets that can be used for detailed 3D building reconstruction. Despite the possibility of using the data presented in Table 5, it is necessary to approach them critically and verify that they are appropriate for the proposed application, or whether there is a need to create an entirely new dataset.

- *Computing power and data bottlenecks:* The variety of applications utilising 3D building models is leading to an increasing demand for data at different scales, from global to local. This, in turn, is associated with an increase in the amount of data that needs to be maintained, transmitted and processed. Although sophisticated methods for building reconstruction may produce more desirable results, the time associated with processing large volumes of data is typically greater than that of less complex techniques, subsequently reducing their cost-effectiveness.

  The solution to the big data processing problem may lie with high-performance computing (HPC) technologies, a set of tools and systems that provide significantly higher performance than a typical desktop computer, laptop, or workstation. HPC can be used to overcome the issue of slow data processing on a local system or limitations of CPU capacity. However, since HPC is often chargeable, a common method is to run a small-scale test on a local device and then migrate the analysis to HPC [162]. Ultimately, coping with big data may contribute to the development of machine-learning algorithms and the search for efficient data storage of pipelines [163,164].

**Table 5.** List of the benchmark datasets from mobile (t.m.) and static (t.s.) terrestrial platforms with labels. For data from multimodal technology, the annotations do not specify building openings. Information about building openings is distinguished in the datasets from the single-sensor technology.

| | Platform | Position (GPS,IMU) | LiDAR | Cameras | Building | Openings | Objects |
|---|---|---|---|---|---|---|---|
| **Multimodal sensor** | | | | | | | |
| **A2D2** Geyer et al. [165] | t.m. | Yes | Yes | Yes | Yes | No | 3D, 2D |
| **KITTI** Geiger et al. [166] | t.m. | Yes | Yes | Yes | Yes | No | 3D, 2D |
| **Apollo Space** Huang et al. [167] | t.m. | Yes | Yes | Yes | Yes | No | 3D, 2D |
| **Mapilliary** Neuhold et al. [168] | t.m. | Yes | Yes | Yes | Yes | No | 3D, 2D |
| **CityScape** Cordts et al. [169] | t.m. | Yes | Yes | Yes | Yes | No | 3D, 2D |
| **TUM-FACADE** [158] | t.m. | No | Yes | No | Yes | Yes | 3D |
| **Semantic3D** [170] | t.s. | Yes | Yes | No | Yes | Yes | 3D |
| **IDD: India Driving Dataset** Varma et al. [171] | t.m. | Yes | Yes | Yes | Yes | No | 3D,2D |
| **Single sensor** | | | | | | | |
| **eTRIMS** Korč and Förstner [172] | t.s. | No | No | Yes | Yes | Yes | 2D |
| **RueMonge** Riemenschneider et al. [173] | t.s. | No | No | Yes | Yes | Yes | 2D |
| **CMP** Tyleček and Šára [174] | t.s. | No | No | Yes | Yes | Yes | 2D |
| **BDD100K** Yu at al. [175] | t.m. | Yes | No | Yes | Yes | No | 2D |

- *Completeness:* 3D building reconstruction from images is a reverse-engineering task that requires methods to validate newly produced results. One of the metrics for estimating

the quality of the obtained 3D building model is completeness. Completeness has two kinds of errors: omission and commission. The first relates to omitted building elements, while the second concerns elements incorrectly classified as part of the building.

Assessing the completeness of 3D building models generated from terrestrial images is a challenging task due to elements that obscure the façade of the building (e.g., cars, people and vegetation) and a lack of reference data. Although blueprints and BIM can serve as a reference to assess the surface-based performance of developed algorithms, the available BIM and blueprints data are again limited. Therefore, it is essential to develop methods to readily determine the completeness of detailed building models.

## 9. Conclusions

Detailed façade segmentation, which includes the detection and location of building openings, is becoming an essential topic among researchers and industry professionals. However, despite the growing interest in more accurate building modelling, the detection of façade elements still needs to be investigated, and many of the challenges require further attention. In this review, we provided a comprehensive overview of the techniques and sensors used to acquire data to detect and reconstruct building façades and their openings.

The benefits, limitations and research gaps in the field of detailed 3D building reconstruction with a focus on opening extraction have been discussed. The main limitations of the current methods include the problems with the automation of processes that require user intervention and pre- and post-processing. User intervention throughout the entire workflow also influences the final product's quality, indicating the need for operator guidance.

Finally, the constraints associated with obtaining complete data representing buildings pose a challenge to the detailed reconstruction of buildings. Through assessment of the state of the art, the presented overview allows conclusions to be drawn about future research directions. The first is data fusion from multi-modal sources and different platforms to minimize the obscuring of façade elements by static and dynamic elements of the urban environment.

In addition, there is a need for more diverse sets of comparative data that would increase the possibility of creating robust and reproducible solutions for detailed 3D building reconstruction. Another potential direction for future research is the adaptation of computer vision, machine-learning and deep-learning methods to automate the process and limit user interference.

Despite the growing interest in using these strategies, well-established, proven and publicly available solutions are lacking. Finally, there is a need to establish quality matrices to help evaluate the proposed strategies and to develop pipelines for completeness, effectiveness and efficiency.

**Author Contributions:** Conceptualization, A.K., S.C., R.L. and S.G.; methodology, A.K.; formal analysis, A.K.; investigation, A.K.; writing—original draft preparation, A.K; writing—review and editing, S.C., R.L. and S.G.; visualization, A.K.; supervision, S.C., R.L. and S.G. All authors have read and agreed to the published version of the manuscript.

**Funding:** This work was supported by the Engineering and Physical Sciences Research Council Centre for Doctoral Training in Geospatial Systems (grant number EP/S023577/1) with the support of the Ordnance Survey.

**Data Availability Statement:** No new data were created during this study.

**Conflicts of Interest:** The authors declare no conflict of interest.

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
