# Peer review of "Detailed Three-Dimensional Building Façade Reconstruction: A Review on Applications, Data and Technologies"

_remotesensing, doi:10.3390/rs14112579_

Round 1

Reviewer 1 Report

overall, the authors did a hard work to review recent progress on detailed 3D building facade reconstruction, it's valuable for the academia and industry. However, there still exists some major issues need to be resolved. First of all, it is suggested to add an overall framework of fundamental theories, revelant technologies, and applications, which would provide a big picture for the readers. What's more, instead of listing what has been done by whom, please discuss and compare their advantages and disadvantages in solving real-world problems, then highlight potential research gaps for furture works. In addition, evidences on how the conclusions are made needs more elaborations. some comments are: 

  1. the title should be revised to keep consistent with the content, i.e., change to "detailed three-dimensional building facades reconstruction: a review on data, technologies and applications"
  2. does section 3 needed for this paper, if the answer is yes, please elaborate more on the reasons
  3. please provide indicators or creteria used to classify different literatures and from what aspects do you compare them and why? listing a table if possible
  4. when comparing different works, the fundamental techniques they used, what constraints & datasources, for what scenarios, and potential limitations, etc., may be considered
  5. due to the above issues, how the limitations and challenges are obtained is not clear now, and more discussions and evidences are needed.

Reviewer 2 Report

This paper presents a summary of data, technologies, and applications about detailed three-dimensional building façade. However, there are some limitations:

(1) The manuscript could benefit from more structure (perhaps a table of contents) or an extended "highlights" section/figure/table summarizing key findings/recommendations. There is a lot of very useful information here, but at times it can be difficult to parse through in the sense that key messages can be lost in the broader context of information. I find the tables with examples (e.g. Table 1) included in the manuscript useful to summarize some of these key pieces of information, but I think that because this paper is so dense, it could also benefit from some sort of table of contents/outline presented upfront. It could help researchers/readers find key pieces of information relevant to their work.

(2) Other review paper discussed similar topic should be presented in the Introduction section.

(3) All potential applications of detailed 3D building faced should be presented and each may serve as a subsection for 2. Applications.

(4) Why are platforms and Data types/representation included in the section—fundamental technologies?

(5) Is discussion about 2D scene interpretation necessary since our topic is about 3D building façade interpretation?

(6) Some typo errors: man¬aging, Othershttps:, Figure2), topic. existing research, and t.m (Table 2).

(7) No ‘object extraction’ in Figure 2. LoD3 is not explained here until Section 3. Levels of detail.
(8) t.m. and t.s. are not clear.

Reviewer 3 Report

Good effort with the following notes:

  • some figures have very small font.
  • some aspects of the sections go into significant depth versus others - this is disrupting the flow of the story and imbalances the overview. Perhaps some resources should be eliminated in favor of others to emphasize particular directions. Such work can hardly be comprehensive and informative at the same time.
  • majority of figures are taken from other authors - the issue is what made these particular figures important enough to be included in this review? Would recommend re-evaluation of figures and their contribution to the discussion at hand.
  • some sections go decades back - while foundations are important, how are these applicable to newer applications/implementations? Would recommend re-evaluation of some resources and theories in favor of more recent applications and/or implementations.

Reviewer 4 Report

The review paper is very well written and incorporates the latest research in the field. The authors provide an overview of the emerging acquisition and processing techniques for building façade reconstruction from terrestrial and aerial data, emphasizing on building opening detection. Definitely will be of interest to the readers and scientific community. I have provided more detailed comments in an attached file. My main concerns and suggestions are related to the abstract and the reordering of the paper. The technologies should be filled by the applications.

Reviewer 5 Report

This submission reviews the literature about sensor-enabled automatic 3D reconstruction (representation) of buildings on a city scale, focusing on techniques and approaches to modeling of building fenestration. The manuscript is overall well organized and thus can be considered for further process of publication. That said, careful revision is also necessary to improve the quality of this work. Please find some comments below:

  • Add your findings (key technical methods, advantages, limitations in current research, etc.) briefly in the abstract. 
  • Add a "method" section that explains your own method(s) regarding collection of data/bibliographic sources and analysis. A graphic that shows a yearly trend of publication may also be necessary.
  • Make at least two or more tables that summarize and compare previous studies in terms of the point of your interest. 

Round 2

Reviewer 1 Report

accept as is